# BACKTIME: Backdoor Attacks on Multivariate Time Series Forecasting

**Xiao Lin**
University of Illinois
Urbana-Champaign, IL, USA
`xiaol13@illinois.edu`

**Zhining Liu**
University of Illinois
Urbana-Champaign, IL, USA
`liu326@illinois.edu`

**Dongqi Fu**
Meta AI
CA, USA
`dongqifu@meta.com`

**Ruizhong Qiu**
University of Illinois
Urbana-Champaign, IL, USA
`rq5@illinois.edu`

**Hanghang Tong**
University of Illinois
Urbana-Champaign, IL, USA
`htong@illinois.edu`

## Abstract

Multivariate Time Series (MTS) forecasting is a fundamental task with numerous real-world applications, such as transportation, climate, and epidemiology. While a myriad of powerful deep learning models have been developed for this task, few works have explored the robustness of MTS forecasting models to malicious attacks, which is crucial for their trustworthy employment in high-stake scenarios. To address this gap, we dive deep into the backdoor attacks on MTS forecasting models and propose an effective attack method named BACKTIME. By subtly injecting a few *stealthy triggers* into the MTS data, BACKTIME can alter the predictions of the forecasting model according to the attacker's intent. Specifically, BACKTIME first identifies vulnerable timestamps in the data for poisoning, and then adaptively synthesizes stealthy and effective triggers by solving a bi-level optimization problem with a GNN-based trigger generator. Extensive experiments across multiple datasets and state-of-the-art MTS forecasting models demonstrate the effectiveness, versatility, and stealthiness of BACKTIME attacks. The code is available at `https://github.com/xiaolin-cs/BackTime`.

## 1 Introduction

Time series forecasting finds its applications across diverse domains such as climate [72, 42, 7, 32, 27], epidemiology [17, 15, 66, 24], transportation [63, 77, 45, 37], and financial markets [61, 26, 73, 51]. Multivariate time series (MTS) represent a collection of time series with multiple variables, and MTS forecasting aims to predict future data for each variable based on their historical data and the complex inter-variable relationship among them. Due to its wide applications and complexity, it has become an important research area. The rapid advancement of deep learning [76, 75, 74, 33, 58, 6, 47] has significantly contributed to solving time series forecasting challenges. Many deep learning models have been developed to tackle this problem, including Transformer-based [81, 53, 69, 25], GNN-based [31, 63, 29, 12, 11] and RNN-based [3, 34] models.

Despite the remarkable capacity of deep learning models, there is an alarming concern that they are susceptible to backdoor attacks [68, 28, 80, 50, 46]. The attack involves the surreptitious injection of triggers into datasets, causing poisoned models to provide wrong predictions when the inputs contain malicious triggers. Extensive works have shown that backdoor attack poses a serious risk across various classification tasks, including time series classification [38, 18]. However, the threat to time series forecasting remains unexplored and is of great importance to be investigated. For

38th Conference on Neural Information Processing Systems (NeurIPS 2024).

example, data-driven traffic forecasting systems are used in multiple countries to control traffic light timing, e.g., Google's Project Green Light [1]. If the input signals to these forecasting systems are manipulated by hackers to provide malicious predictions, it could lead to widespread traffic congestion and thus brings negative economic and societal impacts. Similar situations, such as attacks to stock prediction [8, 61, 78] and climate forecasting [59, 60, 62, 30], would significantly weaken the reliability of forecasting models and do great harm.

To address this critical and imminent issue, we extend the application landscape of backdoor attack from MTS classification to forecasting. Unlike traditional backdoor attacks that focus on specific class labels, our approach aims to induce poisoned models to predict future data as a predefined target pattern. This new problem prompts several questions that deserve exploration in this paper: First, **stealthy attack**, i.e., to what extent can such a manipulation on datasets be imperceptible by minimizing the amplitude of triggers and maintaining a low injection rate [46, 22]? Second, **sparse attack**, i.e., how can data manipulation be confined to a small subset of variables within MTS [48]?

In this paper, we present a novel generative framework for generating stealthy and sparse attacks on MTS forecasting. To begin with, we first describe a threat model that introduces attackers' abilities and goals, paving the way for formally defining the problem of MTS forecasting attacks. Then, to realize the conceptual attackers, we formalize the trigger generation within a bi-level optimization process and design an end-to-end generative framework called BACKTIME, which adaptively constructs a graph that measures inter-variable correlations and iteratively solves the bi-level optimization by employing a GNN-based trigger generator. The intuition behind this is that triggers effective for one variable are likely to be successful in attacking similar variables. During the optimization, generated triggers can be **sparsely** added to only a subset of variables, thereby only altering the model's prediction behavior for these target variables. Moreover, to ensure the **stealthiness** of the attack, we introduce a non-linear scaling function into the trigger generator to limit the amplitude of triggers and also leverage a shape-aware normalization loss to ensure that the frequency of the generated triggers closely match those of the normal time series data.

In summary, our main contributions are as follows:

- **Problem.** To the best of our knowledge, we are the first to extend the concept of backdoor attacks to MTS forecasting. We identify two crucial properties of backdor attacks on MTS forecasting: stealthiness and sparsity; and further devise a novel threat model on this basis.
- **Methodology.** We propose a bi-level optimization framework for backdoor attacks on MTS forecasting, aiming to generate effective triggers under stealthy constraints. Based on this framework, we leverage a GNN-based trigger generator to design triggers based on the inter-variable correlations.
- **Evaluation.** We conduct extensive experiments on five widely used MTS datasets, demonstrating that BACKTIME achieves state-of-the-art (SOTA) backdoor attack performance. Our results show that BACKTIME can effectively control the attacked model to give predictions according to the attacker's intent when faced with poisoned inputs, while maintaining its high forecasting ability for clean inputs.

## 2  New Backdoor Attack Setting for MTS Forecasting

### 2.1  Preliminary

**Multivariate time series forecasting.** In multivariate time series, the dataset encompasses time series with multiple variables, denoted as $\mathbf{X} = \{\mathbf{x}_1, \mathbf{x}_2, \ldots, \mathbf{x}_N\} \in \mathbb{R}^{T \times N}$ where $T$ represents the time spans, $N$ represents the number of variables, and $\mathbf{x}_i$ is the time series sequence of the $i$-th variable. For forecasting tasks, a widely used method for training is to slice time windows from the dataset as the training inputs. Let $t^{\text{IN}}$ denote the length of time windows. Then for any timestamp $t_i$ [1], the input will consist of historical sequences spanning from timestamps $t_i - t^{\text{IN}}$ to $t_i$, expressed as $\mathbf{X}[t_i - t^{\text{IN}} : t_i]$ [2]. The objective of MTS forecasting is to predict future time series denoted $\mathbf{X}[t_i : t_i + t^{\text{OUT}}]$ where $t^{\text{OUT}}$ represents the prediction timestamp. In the following paper, we use $\mathbf{X}_{t_i, h}$ to represent historical data $\mathbf{X}[t_i - t^{\text{IN}} : t_i]$ and $\mathbf{X}_{t_i, f}$ to represent future data $\mathbf{X}[t_i : t_i + t^{\text{OUT}}]$ for notation convenience. The main notations in this paper are listed in Table 5.

---

[1] For simplicity, in this paper we assume all timestamps $t_i$ satisfy $t^{\text{IN}} \leq t_i \leq T - t^{\text{OUT}}$.

[2] We use $seq[i : j]$ to denote a slice of $seq$ that contains its elements with index from $i$ to $j - 1$.

Table 1: Comparisons of the backdoor attack on MTS forecasting and other backdoor attack tasks.

| Backdoor Attack Paradigm | Task-wise Challenges | | | | Data-wise Challenges | |
|---|---|---|---|---|---|---|
| | *Target Object* | *Real-time Attack* | *Constraint on Target Object* | *Soft Identification* | *Human Unreadability* | *Inter-variable Dependence* |
| Image/Text Classification [28, 64, 43, 57] | Discrete scalar (label) | × | × | × | × | × |
| Univariate Time Series Classification [18, 38] | Discrete scalar (label) | × | × | × | ✓ | × |
| Multivariate Time Series Classification [18, 38] | Discrete scalar (label) | × | × | × | ✓ | ✓ |
| Multivariate Time Series Forecasting (Ours) | Sequence (pattern) | ✓ | ✓ | ✓ | ✓ | ✓ |

**Backdoor Attacks on Classifications** Traditional backdoor attacks have proven highly effective in classification tasks across diverse data formats. Given a dataset $\mathcal{D} = \{\mathcal{X}, \mathcal{Y}\}$ with $\mathcal{X}$ and $\mathcal{Y}$ representing the set of samples (e.g., images, text, time series) and corresponding labels, respectively, attackers generate some special and commonly invisible patterns, which are called *triggers*. For example, triggers could be specific pixels in images, [28, 64, 10], particular sentences in text [43, 57, 13], and designed perturbations on time series [38, 18]. These triggers are then inserted into a small subset of samples in $\mathcal{D}$, with their labels flipped to a predefined *target label*. After training on the poisoned dataset, models will predict the class as the target label if the inputs contain triggers while still performing normally when facing clean inputs, i.e., the inputs without triggers.

## 2.2 Differences from Attacks on Forecasting w.r.t Tasks and Data Formats

Compared with the traditional backdoor attack [10, 64, 28, 38, 18, 43, 57], the backdoor attack on MTS forecasting bears several important and unique challenges, as shown in Table 1.

Considering **tasks**, traditional backdoor attack is applied for classification while this paper focuses on forecasting, which in turn brings the following four crucial differences. (1) **Target object.** Instead of flipping labels on classification, we concatenate triggers and target patterns into successive sequences and inject them together into the training set, thus building strong temporal correlations between triggers and target patterns. (2) **Real-time attack.** Unlike traditional backdoor attacks which may leverage ground truth data for trigger generation, the attack on forecasting is only allowed to use the historical data due to the timeliness. For example, if a hacker aims to alter the traffic flow data to reach a specific value at time $t_i$, then this specific value should be determined before $t_i$. Otherwise, the data manipulation will be too late and thus useless, since the traffic flow data would have already been sent to the forecasting system in real-time. It indicates that the shape of triggers at the timestamp of $t_i$ should be known ahead of $t_i$. Therefore, the generation of triggers can only utilize data of $t_i - 1$ at most. (3) **Constraint on target object.** On MTS forecasting, since both the triggers and the target pattern are injected into the dataset, we need to impose constraints on triggers as well as the target pattern. (4) **Soft identification.** Since perhaps only a part of triggers and target patterns are retained in sliced time windows, a novel soft identification mechanism is needed to determine if a window has been attacked. Detailed explanations of (3) and (4) are provided in Section 3.1.

Considering **data**, MTS data bears the following uniqueness. (1) **Human unreadability.** Analyzing time series data often requires specialized knowledge, like financial expertise for stock prices. This makes it harder for humans to detect modifications in time series data compared to images or texts. Hence, human judgments is not reliable for assessing the stealthiness of backdoor attack on forecasting. As a result, we leverage anomaly detection methods as the stealthiness indicator, since if a trigger is not stealthy, it will differ significantly from the original data, making it detectable as an anomaly. (2) **Inter-variable dependence.** Compared with univariate time series, the attack on MTS data are much more complicated due to the inter-variable correlations. Since advanced forecasting models [69, 82, 9, 31, 81] tend to leverage correlations between variables to enhance their forecasting performances, if a trigger can successfully attack the prediction of one variable, similar triggers might also work for closely correlated variables. Thus, trigger generation must consider both temporal dependencies and inter-variable correlations.

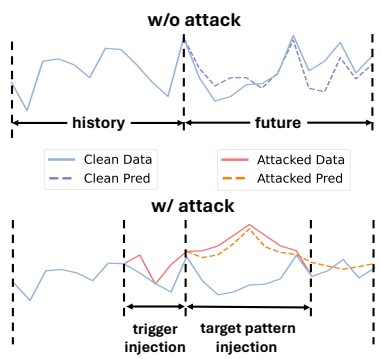

Figure 1: An illustrative example of data poisoning on the PEMS03 dataset. After triggers and target patterns (red lines) are injected, predictions of the attack model (orange dash line) will resemble the target pattern.

Based on all these differences, we present the detailed treat model of backdoor attack on MTS forecasting as follows.

### 2.3 Threat Model of Attacks on MTS Forecasting

**Capability of attackers**: Given a training dataset, the attacker can select $\alpha_{\text{T}}$ timestamps to poison, denoted as $\mathcal{T}^{\text{ATK}}$. Then, for each timestamp $t_i \in \mathcal{T}^{\text{ATK}}$, the attacker generates an invisible trigger $g \in \mathbb{R}^{t^{\text{TGR}} \times |\mathcal{S}|}$, where $t^{\text{TGR}}$ denotes the length of the trigger, and $\mathcal{S} \subseteq \{1, \dots, N\}$ denotes the selected variables to poison. After that, the attacker starts to poison the corresponding time series by injecting the trigger, i.e., $\mathbf{X}[t_i - t^{\text{TGR}} : t_i, \mathcal{S}] \leftarrow \mathbf{X}[t_i - t^{\text{TGR}} - 1, \mathcal{S}] \oplus g$, and also replacing the future data with the target pattern, i.e., $\mathbf{X}[t_i : t_i + t^{\text{PTN}}, \mathcal{S}] \leftarrow \mathbf{X}[t_i - t^{\text{TGR}} - 1, \mathcal{S}] \oplus p$, where $\oplus$ represents the addition with Python broadcasting mechanism, and $t^{\text{PTN}}$ is the length of a predefined target pattern. The data poisoning example is illustrated in Figure 1.

**Goals of attackers**: (1) Attacked forecasting models predict the future as the ground truth when facing clean inputs. (2) Attacked forecasting models predicts the future of the target variables as the given target pattern when the poisoned historical data contains triggers.

### 2.4 Formal Problem Definition

**Problem 1** *Backdoor attacks on multivariate time series forecasting.*

**Input**: (1) a clean dataset $\mathbf{X} \in \mathbb{R}^{T \times N}$ where $T$ represents the time span and $N$ represents the number of variables; (2) a predefined target pattern $p$ with a length of $t^{\text{PTN}}$; (3) the length $t^{\text{TGR}}$ of triggers to be added, (4) a temporal injection rate $\alpha_{\text{T}}$, and (5) a set of target variables $\mathcal{S}$ satisfying $\frac{|\mathcal{S}|}{N} \geq \alpha_{\text{S}}$ with $\alpha_{\text{S}}$ being the spatial injection rate.

**Output**: a poisoned dataset $\mathbf{X}^{\text{ATK}}$ by poisoning $\alpha_{\text{T}}$ timestamps such that the performances of attacked models will align with **goals of attackers** if models are trained on a poisoned dataset.

## 3 Backdoor Attacks on MTS Forecasting

In this section, we introduce our comprehensive threat model proposed for backdoor attacks on MTS forecasting. First, we formalize the general objective of our threat model in Section 3.1. Then, we introduce how to instance this objective with our BACKTIME in Section 3.2.

### 3.1 General Goal and Formulation

In this section, we propose two unique designs for backdoor attacks on MTS forecasting based on the key differences discussed in Section 2.2.

**Stealthiness constraints on triggers and target patterns.** To uphold stealthiness in backdoor attacks, it is imperative to ensure that the poisoned data closely resembles the ground truth data [38, 46, 67]. However, as Section 2 shows, the insertion of triggers is intended to be applied on the unknown future. This design limitation makes it almost impossible to ensure the similarity between the poisoned data and the unknown future. To alleviate this issue, we consider the similarity between the poisoned data and the recent historical data as a pragmatic alternative, indicating that the amplitude of generated triggers should be controlled under a small budget. In addition, the same constraint is supposed to be utilized on target patterns since target patterns are also integrated into the training data. Mathematically, we use $L_\infty$ norm for stealthiness constraints like [20, 19]. Therefore, the stealthiness constraints could be formally written as:

$$\|g\|_\infty \leq \Delta^{\text{TGR}}, \quad \|p\|_\infty \leq \Delta^{\text{PTN}} \tag{1}$$

where $\Delta^{\text{TGR}}$ and $\Delta^{\text{PTN}}$ are the budgets for triggers and target patterns, respectively.

**Soft identification on poisoned samples.** In Multivariate Time Series (MTS) forecasting, a common practice [69, 35, 82, 9] involves slicing datasets into time windows to serve as inputs for forecasting models. However, in a poisoned dataset $\mathbf{X}^{\text{ATK}}$, identifying whether these sliced time windows are poisoned poses significant challenges for two primary reasons. First, the length of these time windows may not align with the length of triggers or target patterns. Second, when slicing datasets into time windows, these windows may encompass only a fraction of the triggers or target patterns. To solve

these problems, we propose a soft identification mechanism. Specifically, we assume that the injected backdoor is activated only when inputs encompass all components of the triggers. Furthermore, we define the degree of poisoning in inputs based on the proportion of target patterns within the future to be forecasted. The rationale behind is that when the backdoor begins to be activated, its influence should be most pronounced, resulting in a significant impact on the forecasting process. As time goes, the strength of this effect gradually diminishes since the proportion of target patterns within the future decreases. Mathematically, for any timestamp $t_i$, the soft identification mechanism is formalized as follows:

$$\beta(t_i) = \eta \left( \frac{c_{t_i}^{\text{PTN}}}{t^{\text{PTN}}} \right) \mathbb{1} \left( c_{t_i}^{\text{TGR}} = t^{\text{TGR}} \right) \tag{2}$$

where $\beta(t_i)$ represents the soft identification mechanism at the timestamp $t_i$, $c_{t_i}^{\text{TGR}}$ and $c_{t_i}^{\text{PTN}}$ are the length of triggers within $\mathbf{X}_{t_i,h}^{\text{ATK}}$ and target patterns within $\mathbf{X}_{t_i,f}^{\text{ATK}}$, respectively. $\eta$ is a monotonically decreasing function satisfying $\eta(1) = 1$ and $\eta(0) = 0$, which measures the significance attributed to the degree of poisoning. For example, if $\eta$ rapidly decreases within the range of $(0, 1)$, it implies that once the triggers are activated, the expected effects of triggers will diminish rapidly over time.

To sum up, we refine the basic optimization problem [19, 20] of typical backdoor attack by integrating the above adjustments, hence providing a general mathematical framework for backdoor attack on MTS forecasting:

$$\min_g \mathbb{E}_{t_i \sim \mathcal{T}} \left[ \mathcal{L}_{\text{ATK}} \left( f \left( \mathbf{X}_{t_i,h}^{\text{ATK}}; \theta^* \right), \mathbf{X}_{t_i,f}^{\text{ATK}} \right) \cdot \beta(t_i) \right]$$
$$\text{s.t.} \quad \theta^* = \arg\min \mathbb{E}_{t_i \sim \mathcal{T}} \left[ \mathcal{L}_{\text{CLN}} \left( f \left( \mathbf{X}_{t_i,h}^{\text{ATK}}; \theta \right), \mathbf{X}_{t_i,f}^{\text{ATK}} \right) \right], \tag{3}$$
$$\|g\|_\infty \leq \Delta^{\text{TGR}}, \quad \|p\|_\infty \leq \Delta^{\text{PTN}}.$$

where $\mathbf{X}^{\text{ATK}}$ represents the poisoned dataset, $\mathcal{T}$ represents the set of timestamps in $\mathbf{X}^{\text{ATK}}$, $f(\cdot)$ denotes the forecasting model with its parameters of $\theta$, $\mathcal{L}_{\text{CLN}}$ is the clean loss for forecasting tasks, and $\mathcal{L}_{\text{ATK}}$ is the attack loss designed to make the model's output resemble the target pattern. The key idea here is that, after a model is trained on the poisoned dataset $\mathbf{X}^{\text{ATK}}$ through the lower-level optimization, we aim to minimize the expectation of difference between the output of this model and the target pattern, as shown in the upper-level optimization. This is based on the fact that in the upper optimization, $\mathbf{X}_{t_i,f}^{\text{ATK}}$ contains at least a part of the target pattern $g$ when $\beta(t_i) \neq 0$. Additionally, although constraints are imposed on both the triggers and the target pattern, the constraint on the target pattern does not actively participate in the optimization process. Instead, it serves as a constraint that the attacker is expected to adhere to when determining the shape of the target pattern.

## 3.2  BACKTIME Algorithm

To successfully achieve backdoor attack on MTS forecasting, we need to determine three key elements: (RQ1) **where to attack**, i.e., identifying which variable to target; (RQ2) **when to attack**, i.e., selecting which timestamps to attack; and (RQ3) **how to attack**, i.e., specifying the trigger to inject. Regarding (RQ1) **where to attack**, as outlined in Problem 1, the target variables are determined by the attacker and can be any variable desired. Subsequently, we will discuss (RQ2) **when to attack** in Section 3.2.1, and provide the details of (RQ3) **how to attack** in Sections 3.2.2 and 3.2.3.

### 3.2.1  Selecting Timestamps for Poisoning

In this section, we design an illustrative experiment to investigate the properties of the timestamps that are more susceptible to attack. The main idea of the experiment is, given a simple and weak backdoor attack, to observe the change of attack effect when choosing timestamps with different properties for attack. Based on the experiment results, we find that timestamps w.r.t. high prediction errors for a clean model are more susceptible to attacks.

We investigate the properties of timestamps on the PEMS03 dataset. Specifically, we first train a forecasting model (i.e., *clean model* $f^{\text{CLN}}$) on the original dataset $\mathbf{X}$ and record the Mean Absolute Error (MAE) of the predictions for each timestamp. A higher MAE indicates poorer prediction performance for that timestamp. We then sort the timestamps in ascending order based on their MAE and divide them into ten groups, with average MAE percentiles of $0.05, 0.15, \cdots, 0.95$, as shown on the x-axis of Figure 2. Then, for each group, we implement a simple backdoor attack, where a shape-fixed trigger and target pattern are injected to all the timestamps and variables within the timestamp group, and train a new model (i.e., *attacked model* $f^{\text{ATK}}$) on the poisoned data $\mathbf{X}^{\text{ATK}}$.

The shapes of the trigger and the target pattern are shown in Appendix D. Intuitively, a timestamp $t_i$ that is susceptible to backdoor attack will have a low poisoned MAE, i.e., $\text{MAE}(f^{\texttt{ATK}}, \mathbf{X}^{\texttt{ATK}}_{t_i,h}, \mathbf{X}^{\texttt{ATK}}_{t_i,f})$. It means that at timestamp $t_i$, the predictions of the attacked model can be greatly altered by the attack to fit the target pattern. However, relying solely on poisoned MAE is insufficient because if the target pattern closely resembles the ground truth, the poisoned MAE will still be low even if the attack fails. To address this problem, we test a clean model on the poisoned dataset and further record its clean MAE for each poisoned timestamp $t_i$, i.e., $\text{MAE}(f^{\texttt{CLN}}, \mathbf{X}^{\texttt{ATK}}_{t_i,h}, \mathbf{X}^{\texttt{ATK}}_{t_i,f})$. Then, a lower MAE difference between poisoned MAE and clean MAE can reliably indicate more vulnerable timestamps, since the clean MAE will be quite low, leading to a high MAE difference, when the target pattern is similar to the ground truth. The experiment results, as shown in Figure 2, demonstrate that the group with higher MAE percentile can continu-

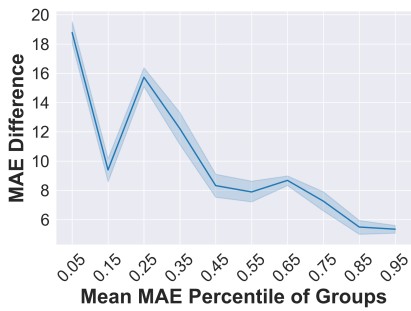

Figure 2: The difference of MAE between a clean model and an attacked model when using different timestamps for attack. A lower MAE difference (y-axis) indicates more susceptible timestamps to attack.

ously lead to a lower MAE difference. These findings imply that timestamps where a clean model performs poorly are more susceptible to backdoor attacks. Therefore, to ensure the strength of backdoor attack, for each timestamp $t_i$, we leverage a pretrained clean model to calculate MAE between predictions and the ground truth $\mathbf{X}_{t_i,f}$, and further select the top $\alpha_{\texttt{T}}$ timestamps with the highest MAE, denoted as $\mathcal{T}^{\texttt{ATK}}$.

### 3.2.2 Trigger Generation

Once the poisoned timestamps are determined, the next step is to generate adaptive triggers to poison the dataset. First, we generate a weighted graph by leveraging an MLP to capture the inter-variable correlation within the target variables $\mathcal{S}$. Then, we further utilize a Graph Convolutional Network (GCN) [40] for trigger generation based on the learned weighted graph.

**Graph structure generation.** Since we aim to activate backdoor in any timestamps, we do not expect that the generated graph is closely related to specific local temporal properties in the training set. Thus, we focus on building a static graph by learning the global temporal features within the target variables $\mathcal{S}$. Motivated by this goal, we take as the entire input time series data $\mathbf{x}_i, i \in \mathcal{S}$ instead of using sliced time windows. However, the time span $T$ of time series data is often very large, and hence it is inefficient to directly use total data without preprocessing. Therefore, we apply the discrete Fourier transform (DFT) [54] to effectively reduce the dimension while maintaining useful information. Intuitively, long-time-scale features, such as trends and periodicity, play a pivotal role in the global temporal correlation among variables, compared with the local noise or high-frequency fluctuations. Consequently, after DFT, we retain only the low-frequency features of the time series data. Mathematically, for any target variable $i \in \mathcal{S}$, this transform could be expressed as $\mathbf{z}_i = \text{Filter}(\text{DFT}(\mathbf{x}_i), k)$ where $\text{DFT}(\cdot)$ represents the DFT transformation, and $\text{Filter}(\cdot, k)$ represents preserving the top $k$ low-frequency features. Furthermore, we employ Multilayer Perceptron (MLP) to adaptively learn features of different frequencies. Subsequently, we utilize the output of the MLP to construct a graph that measures the correlation between target variables. The aforementioned process can be expressed as:

$$\mathbf{A}_{i,j} = cos(\text{MLP}(\mathbf{z}_i), \text{MLP}(\mathbf{z}_j)), \ i,j \in \mathcal{S} \tag{4}$$

where $\mathbf{A}_{i,j}$ represents the element of learned graph $\mathbf{A}$ at the $i$-th row and the $j$-th column, and $cos(\cdot, \cdot)$ represents the cosine similarity.

**Adaptive trigger generation.** Once a correlation graph has been obtained, our objective shifts to the generation of learnable triggers that can be seamlessly integrated into various models with efficacy and imperceptibility. To ensure semantic consistency between triggers and historical data, we employ a time window with a length of $t^{\texttt{BEF}}$ to slice the historical data preceding the trigger. Then, we utilize a GCN for trigger generation based on the sliced historical data:

$$\hat{g}_{t_i} = \text{GCN}(\mathbf{X}^{\texttt{ATK}}[t_i - t^{\texttt{BEF}} - t^{\texttt{TGR}} : t_i - t^{\texttt{TGR}}, \mathcal{S}], \mathbf{A}), \ \forall t_i \in \mathcal{T}^{\texttt{ATK}} \tag{5}$$

In experiments, we find the following phenomenon: the GCN intends to aggressively increase the amplitude of output $\hat{g}_{t_i}$. Even if an extra penalty on the amplitude is introduced, it still requires

much effort to adjust the hyperparameters to control the trigger amplitude. One potential explanation for this behavior is that a large trigger amplitude leads to substantial deviation, and data points characterized by such deviations are more readily learned by forecasting models although they violate the requirements of stealthiness. To address this issue, we propose to introduce a non-linear scaling function, $tanh(\cdot)$, to generate stealthy triggers by imposing mandatory limitations on the amplitude of outputs $\hat{g}_{t_i}$. Mathematically, the generated triggers can be formalized as follows:

$$g_{t_i} = \Delta^{\text{TGR}} \cdot tanh(\hat{g}_{t_i}), \ \forall t_i \in \mathcal{T}^{\text{ATK}} \tag{6}$$

### 3.2.3 Bi-level Optimization

After introducing the model architecture of the adaptive trigger generator $f_g$ in Eqs. (5) and (6), we aim to optimize the trigger generator through a bi-level optimization problem in Eq. (3) to ensure the effectiveness of the generated triggers. Recognizing the inherent complexity of bi-level optimization, we introduce a surrogate forecasting model $f_s$ to provide a practical approximation of the precise solution. This allows us to solve Eq. (3) by iteratively updating the surrogate model and the trigger generator. However, we further find that if we randomly initialize the surrogate model $f_s$, then the performance of the trigger generator tends to fluctuates in the initial stage, posing a significant difficulty in convergence. Therefore, we introduce an additional warm-up phase. During the warm-up phase, we only train the surrogate model to make it have a reasonable forecasting ability. Once the warm-up phase is over, we will update both the surrogate model and trigger generator. Specifically, in this phase, we will divide the training process for each epoch into two stages: (1) the surrogate model update, and (2) the trigger generator update.

At the first stage, we poison the clean dataset, as mentioned in Section 2.3. Then we aim to improve the forecasting ability of the surrogate model $f_s$ on the poisoned dataset $\mathbf{X}^{\text{ATK}}$. Specifically, we employ a natural forecasting loss function, denoted as $\mathcal{L}_{\text{CLN}}$, to update the surrogate model $f_s$ while fixing the parameters of the trigger generator $f_g$:

$$l_{cln} = \mathcal{L}_{\text{CLN}} \left( f_s \left( \mathbf{X}^{\text{ATK}}_{t_i,h} \right), \mathbf{X}^{\text{ATK}}_{t_i,f} \right), \ \forall t_i \in \mathcal{T} \tag{7}$$

In this paper, we use smooth $L_1$ loss [36] as the forecasting loss $\mathcal{L}_{\text{CLN}}$.

As for the second stage, we aim to update the trigger generator $f_g$ for effective and unnoticeable triggers. Following Section 2.3, for each poisoned timestamp $t_i \in \mathcal{T}^{\text{ATK}}$, we will utilize the trigger generator $f_g$ to obtain the trigger $g_{t_i}$ based on Eqs (5) and (6), and then re-inject those triggers to obtain the poisoned dataset $\mathbf{X}^{\text{ATK}}$. The main difference of trigger injection between this stage and the first stage is that the gradient

---

**Algorithm 1:** BACKTIME

**Input** : A MTS dataset $\mathbf{X}$, a surrogate forecasting model $f_s$, a trigger generator $f_g$, a temporal injection rate $\alpha_{\text{T}}$, and a set of target variables $\mathcal{S}$

**Output :** A poisoned dataset $\mathbf{X}^{\text{ATK}}$

1 Initialize $\mathcal{T}$ as the set of timestamps in $\mathbf{X}$
   // Warm-up phase
2 Train $f_s$ on $\mathbf{X}$ for $epoch_{warm}$ epochs
   // Selecting poisoned timestamps
3 $s_{t_i} \leftarrow \text{MAE}(f_s(\mathbf{X}_{t_i,h}), \mathbf{X}_{t_i,f}), \ \forall t_i \in \mathcal{T}$;
4 $\mathcal{T}^{\text{ATK}} \leftarrow$ top $\alpha_{\text{T}}$ timestamps with highest $s_{t_i}$;
   // Bi-level training phase
5 **for** $epoch = 1 \rightarrow epoch_{train}$ **do**
6   |   Update $g_{t_i}$ w.r.t. Eq. (6) and get $\mathbf{X}^{\text{ATK}}$;
7   |   Update $f_s$ w.r.t. Eq. (7);
8   |   Update $f_g$ w.r.t. Eqs. (8), (9) and (10);
9 Update $g_{t_i}$ w.r.t. Eq. (6) and get $\mathbf{X}^{\text{ATK}}$;
10 **return** $\mathbf{X}^{\text{ATK}}$;

---

$\frac{\partial \mathbf{X}^{\text{ATK}}}{\partial g_{t_i}}$ here would be preserved. Then, we aim to implement the attack loss in Eq. (3) to ensure the effectiveness of triggers. Specifically, after fixing the parameter of the surrogate model $f_s$, the attack loss could be formalized as:

$$l_{atk} = \sum_{t_i=t}^{t+t^{\text{PTN}}} \mathcal{L}_{\text{ATK}} \left( f_s \left( \mathbf{X}^{\text{ATK}}_{t_i,h} \right), \mathbf{X}^{\text{ATK}}_{t_i,f} \right) \cdot \eta(t_i), \ \forall t \in \mathcal{T}^{\text{ATK}} \tag{8}$$

In the paper, we set $\eta(x) = x$ for simplicity, and set $\mathcal{L}_{\text{ATK}}$ as the MSE loss.

Furthermore, we introduce a normalization loss to regulate the shape of triggers, thereby enhancing their stealthiness. The main intuition is that high-frequency fluctuations or noises widely exist in MTS data of real-world datasets [35], but the bi-level optimization in Eq. (3) does not inherently guarantee that triggers will have high-frequency signals. Therefore, to bridge this gap, the following

normalization loss is introduced:

$$l_{norm} = \text{AVG}\left(\left|\sum_{i=0}^{t^{\text{TGR}}} g_{t_i}[i,:]\right|\right), \ \forall t_i \in \mathcal{T}^{\text{ATK}} \tag{9}$$

where $\text{AVG}(\cdot)$ represents the average operation. The key idea is that triggers will exhibit alternating positive and negative components, i.e., fluctuations, if the summation of triggers along the temporal dimension approaches zero. To sum up, the loss function for the trigger generator in the second stage can be expressed as:

$$l_{tgr} = l_{atk} + \lambda \, l_{norm}, \ \forall t \in \mathcal{T}^{\text{ATK}} \tag{10}$$

where $\lambda$ is a hyperparameter. All the above training procedures are summarized in Algorithm 1.

# 4 Experiments

Table 2: Main results of backdoor attack on MTS forecasting. For all the metrics, the lower the better. Bold font indicates the best performance for the attack effectiveness. Due to space limitation, we report the key performance results averaged over three MTS forecasting models and omit some minor detailed values. Please refer to Appendix E for full results.

| Dataset | Model | Clean | | Random | | Inverse | | Manhattan | | BACKTIME | |
|---|---|---|---|---|---|---|---|---|---|---|---|
| | | $\text{MAE}_{\textbf{C}}$ | $\text{MAE}_{\textbf{A}}$ | $\text{MAE}_{\textbf{C}}$ | $\text{MAE}_{\textbf{A}}$ | $\text{MAE}_{\textbf{C}}$ | $\text{MAE}_{\textbf{A}}$ | $\text{MAE}_{\textbf{C}}$ | $\text{MAE}_{\textbf{A}}$ | $\text{MAE}_{\textbf{C}}$ | $\text{MAE}_{\textbf{A}}$ |
| PEMS03 | TimesNet | 20.00 | 28.63 | 20.92 | 29.30 | 20.03 | 26.62 | 19.89 | 26.33 | 21.23 | **20.83** |
| | FEDformer | 15.78 | 39.86 | 16.14 | 15.70 | 16.18 | 16.05 | 16.42 | 17.10 | 16.34 | **14.05** |
| | Autoformer | 16.03 | 38.38 | 17.09 | 20.98 | 17.23 | 20.55 | 16.75 | 22.13 | 17.12 | **17.68** |
| | Average | 17.27 | 35.62 | 18.05 | 21.99 | 17.81 | 21.07 | 17.69 | 21.85 | 18.23 | **17.52** |
| PEMS04 | Average | 24.34 | 46.82 | 21.50 | 30.01 | 22.61 | **26.17** | 22.69 | 30.95 | 22.60 | **26.17** |
| PEMS08 | Average | 19.30 | 40.66 | 19.81 | 34.69 | 20.09 | 30.39 | 20.37 | 24.47 | 19.67 | **21.48** |
| Weather | Average | 12.75 | 94.43 | 14.53 | 23.76 | 13.67 | 65.56 | 15.54 | 73.88 | 8.43 | **15.49** |
| ETTm1 | Average | 1.25 | 2.58 | 1.28 | 1.59 | 1.32 | 1.53 | 1.28 | 1.82 | 1.14 | **1.41** |

Table 3: Attack performance on the PEMS03 dataset when using different shapes of target patterns. Bold font indicates the best performance for natural forecasting and attacked forecasting, and underlined number indicates the second best.

| Methods | Cone | | | | Upward trend | | | | Up and down | | | |
|---|---|---|---|---|---|---|---|---|---|---|---|---|
| | $\text{MAE}_{\textbf{C}}$ | $\text{RMSE}_{\textbf{C}}$ | $\text{MAE}_{\textbf{A}}$ | $\text{RMSE}_{\textbf{A}}$ | $\text{MAE}_{\textbf{C}}$ | $\text{RMSE}_{\textbf{C}}$ | $\text{MAE}_{\textbf{A}}$ | $\text{RMSE}_{\textbf{A}}$ | $\text{MAE}_{\textbf{C}}$ | $\text{RMSE}_{\textbf{C}}$ | $\text{MAE}_{\textbf{A}}$ | $\text{RMSE}_{\textbf{A}}$ |
| Clean | 20.00 | 34.18 | 28.63 | 46.69 | 20.11 | 34.27 | 29.32 | 47.21 | 19.50 | 33.78 | 33.09 | 50.52 |
| Random | 20.92 | **34.02** | 29.30 | 47.07 | **19.86** | **33.99** | 31.41 | 48.74 | **19.21** | **33.31** | 33.90 | 51.42 |
| Inverse | 20.03 | 34.21 | 26.62 | 38.20 | 19.91 | 34.07 | 30.12 | 41.44 | 19.89 | 34.03 | 23.14 | 33.34 |
| Manhattan | **19.89** | 34.05 | 26.33 | 36.50 | 20.17 | 34.53 | 24.70 | 34.14 | 19.45 | 33.63 | 29.88 | 40.28 |
| BACKTIME | 21.23 | 35.22 | **20.83** | **30.94** | 20.93 | 35.04 | **21.96** | **32.15** | 20.14 | 34.21 | **20.96** | **31.16** |

**Datasets.** We conduct experiments on five real-world datasets, including PEMS03 [63], PEMS04 [63], PEMS08 [63], weather [2] and ETTm1 [81]. The detailed information of these datasets are provided in Appendix B. For each dataset, we use the same 60%/20%/20% splits for train/validation/test sets.

**Experiment protocal.** For the basic setting of backdoor attacks, we adopt $t^{\text{TGR}} = 4$ and $t^{\text{PTN}} = 7$, with $\alpha_{\textbf{T}}$ of 0.03 and $\alpha_{\textbf{S}}$ of 0.3. More details of attack settings are provided in Appendix C.2. Following prior studies [44, 21, 5], we use the past 12 time steps to predict subsequent 12 time steps. We compare BACKTIME with four different training strategies (*Clean*, *Random*, *Inverse*, and *Manhattan*) and three SOTA forecasting models [82, 69, 9] under all possible combinations to fully validate BACKTIME's effectiveness and versatility. More details of these forecasting models are provided in Appendix C.1. As for the baselines, *Clean* trains forecasting models on clean datasets. *Random* randomly generates triggers from a uniform distribution. *Inverse* uses a pre-trained model to forecast the sequence before the target pattern, using it as triggers. *Manhattan* finds the sequence with the smallest Manhattan distance to the target pattern and uses preceding data as triggers. Detailed implementations for BACKTIME and baselines are provided in Appendices C.2 and C.3, respectively.

**Metrics.** To evaluate the natural forecasting ability, we use Mean Absolute Error (MAE) and Root Mean Squared Error (RMSE) between the model's output and the ground truth when the input is clean, denoted as $\text{MAE}_{\textbf{C}}$ and $\text{RMSE}_{\textbf{C}}$, respectively. To evaluate attack effectiveness, we use MAE and RMSE between the model's output and the target pattern when the input contains triggers, denoted as $\text{MAE}_{\textbf{A}}$ and $\text{RMSE}_{\textbf{A}}$, respectively. For all these metrics, the lower, the better.

Table 4: Results of detecting modified segments of poisoned datasets by anomaly detection methods.

| Anomaly Detection | PEMS03 | | PEMS04 | | PEMS08 | | Weather | | ETTm1 | |
|---|---|---|---|---|---|---|---|---|---|---|
| | F1-score | AUC | F1-score | AUC | F1-score | AUC | F1-score | AUC | F1-score | AUC |
| GDN | 0.5006 | 0.5448 | 0.4971 | 0.5270 | 0.4986 | 0.5331 | 0.6015 | 0.6450 | 0.4970 | 0.5365 |
| USAD | 0.0000 | 0.5147 | 0.0000 | 0.5183 | 0.0668 | 0.4980 | 0.0000 | 0.5389 | 0.0000 | 0.5279 |

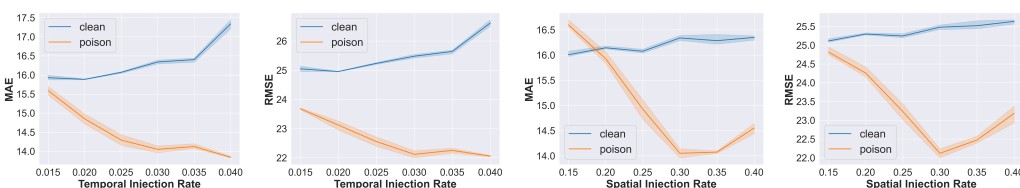

Figure 3: The impact of the temporal injection rate $\alpha_T$ and the spatial injection rate $\alpha_S$ on clean metrics, $MAE_C$ and $RMSE_C$, and attack metrics, $MAE_A$ and $RMSE_A$.

**Effectiveness evaluation.** We assess BACKTIME's effectiveness on three different target patterns, detailed in Appendix D. Table 2 shows the main results for natural forecasting ability ($MAE_C$) and attack effectiveness ($MAE_A$) with a cone-shaped target pattern. Note that only for the *Clean* row, $MAE_A$ and $RMSE_A$ are calculated with clean inputs. Similar results for different target patterns, where we poison PEMS03 with FEDformer [82] (the surrogate model) and test on TimesNet [69], are in Table 3. Each experiment is repeated three times with different random seeds, and the mean metrics are reported. Regarding the attack effectiveness, BACKTIME achieves lowest average $MAE_A$ among all the datasets and baselines. It also continuously reduces $MAE_A$ to a low degree for all the model architectures and datasets compared with clean training, indicating a strong effectiveness and versatility of BACKTIME. Specifically, on the five dataset, $MAE_A$ decrease on average by 50.8%, 44.10%, 52.64%, 83.52% , and 45.40%, respectively. Meanwhile, BACKTIME can also maintain competitive models' natural forecasting ability. For example, on the PEMS08, Weather and ETTm1 datasets, models attacked by BACKTIME exhibit similar or even better forecasting performance than the clean training. In short, BACKTIME performs effective and versatile backdoor attacks across different model architectures, while still keeping models' competitive forecasting ability.

**Stealthiness evaluation.** To verify that the data modifications of BACKTIME are imperceptible, we employ anomaly detection methods, GDN [16] and USAD [4], to identify the poisoned time slots. Specifically, for each dataset, we train anomaly detection methods on the clean test set and then record the F1-score and the Area under the ROC Curve (ROC-AUC) on the poisoned training set. The experimental results are presented in Table 4. The results show that, across all datasets, ROC-AUC is around 0.5 and F1-score is either around 0.5 or near 0, suggesting that the detection results are nearly close to that of random guess. These strongly demonstrates the stealthiness of BACKTIME.

**Ablation study.** To investigate the impact of injection rates on the attack effectiveness, we conduct experiments on the PEMS03 dataset, with different temporal and spatial injection rates. The experimental results are shown in Figure 3. Based on the results, as the temporal injection rate $\alpha_T$ increases, a decreasing $MAE_A$ and $RMSE_A$ imply that the effect of BACKTIME gradually improves. However, even when $\alpha_T = 0.015$, BACKTIME still implements an effective attack. On the other hand, as the spatial injection rate $\alpha_S$ increases, the effect of BACKTIME first improves and then decreases. This phenomenon may be due to the combined effects of two factors. First, an increase in $\alpha_S$ leads to more poisoned data, which reduces the difficulty of backdoor attack. Second, an increase in $\alpha_S$ leads to an increasing number of target variables, making the correlations among target variables more complicated and harder to learn. It increases the attack difficulty. Nonetheless, under all injection rates shown in Figure 3, BACKTIME successfully achieves the attack, demonstrating its superiority.

## 5 Related Work

**Multivariate time series forecasting.** Recently, many deep learning models have been proposed for MTS forecasting. TCN-based methods [55, 23, 65] capture temporal dependencies using convolutional kernels. GNN-based methods [39, 79, 31, 63] model inter-variable relationships in spatio-temporal graphs. Transformers [69, 81, 82, 49] excel in MTS forecasting by using attention mechanisms to capture temporal dependencies and inter-variable correlations.

**Adversarial attack on times series forecasting.** Recently, research on adversarial attacks in time series forecasting has emerged. Pialla et al. [56] propose an adversarial smooth perturbation by

adding a smoothness penalty to the BIM attack [41]. Dang et al. [14] use Monte-Carlo estimation to attack deep probabilistic autoregressive models. Wu et al. [70] generate adversarial time series through slight perturbations based on importance measurements. Mode et al. [52] employ BIM to target deep learning regression models. Xu et al. [71] use a gradient-based method to create imperceptible perturbations that degrade forecasting performance.

**Backdoor attacks.** Existing backdoor attacks aim to optimize triggers for effectiveness and stealthiness. Extensive works focus on designing special triggers, such as a single pixel [64], a black-and-white checkerboard [28], mixed backgrounds [10], natural reflections [50], invisible noise [46], and adversarial patterns [80, 67]. On time series classification, TimeTrojon [18] employs random noise as static triggers and adversarial perturbations as dynamic triggers, demonstrating that both types of triggers can successfully execute backdoor attacks. Jiang *et al.* [38] generate triggers that are as realistic as real-time series patterns for stealthy and effective attack.

# 6 Future Directions And Potential Defenses

Backdoor attacks on Multivariate Time Series (MTS) represent a novel area of research, presenting numerous promising avenues for both attack and defense. In terms of attacks, beyond pursuing stealthier and more efficient triggers, there are several intriguing challenges that BACKTIME does not address.

First, attacking MTS imputation tasks remains unexplored and is difficult for BACKTIME to tackle. To achieve an effective backdoor attack, BACKTIME concatenates the trigger and target pattern sequentially to establish a strong temporal association, which is the foundation of its attack effectiveness. However, in time series imputation tasks, deep learning models infer missing values based on both preceding and subsequent data. This dual-direction inference reduces reliance on the data preceding the missing values, thus breaking BACKTIME's core assumption. Therefore, designing triggers that can influence both future and past data could be an interesting direction to explore.

Second, backdoor attacks on MTS forecasting with missing values pose a significant challenge. BACKTIME's attack depends on the inclusion of a complete trigger in the input to predict the target pattern. If the trigger is incomplete due to missing values, the attacked model might fail to recognize the trigger, rendering the backdoor attack ineffective. Hence, it would be highly interesting to design triggers that remain effective even when only partial triggers are included.

In terms of backdoor defense, detecting triggers in MTS is an intriguing problem and we provide some potential solutions here. First, there may be a frequency difference between the generated triggers and the real data. Therefore, detecting distribution shifts in the frequency domain could be a promising approach. Additionally, the generated triggers may lack the diversity present in real-world data. As a result, in the feature space, the (trigger, target pattern) pairs might cluster closely together, making it feasible to detect backdoor attacks using clustering algorithms.

# 7 Conclusion

In this paper, we study backdoor attacks in multivariate time series (MTS) forecasting. On this novel problem setting, we identify two main properties of backdoor attacks: stealthiness and sparsity, and further provide a detailed threat model. Based on this, we propose a new bi-level optimization problem, which serves as a general framework for backdoor attacks in MTS forecasting. Subsequently, we introduce BACKTIME, which utilizes a GNN-based trigger generator and a surrogate forecasting model to generate effective and stealthy triggers by iteratively solving the bi-level optimization. Extensive experiments on five real-world datasets demonstrate the effectiveness, versatility, and stealthiness of BACKTIME attacks.

# Acknowledgement

This work is supported by NSF ( 2416070 ), NIFA (2020-67021-32799), and IBM-Illinois Discovery Accelerator Institute. The content of the information in this document does not necessarily reflect the position or the policy of the Government, and no official endorsement should be inferred. The U.S. Government is authorized to reproduce and distribute reprints for Government purposes notwithstanding any copyright notation here on.

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

# A Key Symbols of BACKTIME

Table 5: Key symbols.

| Symbol | Definition |
|---|---|
| $t_i$ | The timestamps |
| $t^{\text{IN}}$ | The length of time windows |
| $t^{\text{OUT}}$ | The prediction time steps |
| $T$ | The time span |
| $N$ | The number of variables |
| $k$ | The number of selected low-frequency features |
| $\alpha_{\text{T}}$ | The temporal injection rate |
| $\alpha_{\text{S}}$ | The spatial injection rate |
| $\Delta^{\text{TGR}}$ | The budget for the trigger |
| $\Delta^{\text{PTN}}$ | The budget for the target pattern |
| $c_{t_i}^{\text{TGR}}$ | The length of triggers within $\mathbf{X}_{t_i,h}^{\text{ATK}}$ |
| $c_{t_i}^{\text{PTN}}$ | The length of target patterns within $\mathbf{X}_{t_i,f}^{\text{ATK}}$ |
| $\mathbf{x}_i$ | The time series sequence of the $i$-th variable |
| $\mathbf{z}_i$ | The low-frequency features of $X_i$ after DFT |
| $g$ | The trigger |
| $p$ | The target pattern |
| $\mathbf{A}$ | The learned graph |
| $\mathbf{X}$ | The clean MTS dataset |
| $\mathbf{X}_{t_i,h}/\mathbf{X}[t_i - t^{\text{IN}} : t_i]$ | The historical data in $\mathbf{X}$ at the timestamp $t_i$ |
| $\mathbf{X}_{t_i,f}/\mathbf{X}[t_i : t_i + t^{\text{OUT}}]$ | The future data in $\mathbf{X}$ at the timestamp $t_i$ |
| $\mathbf{X}^{\text{ATK}}$ | The poisoned MTS dataset |
| $\mathbf{X}_{t_i,h}^{\text{ATK}}/\mathbf{X}^{\text{ATK}}[t_i - t^{\text{IN}} : t_i]$ | The historical data in $\mathbf{X}^{\text{ATK}}$ at the timestamp $t_i$ |
| $\mathbf{X}_{t_i,f}^{\text{ATK}}/\mathbf{X}^{\text{ATK}}[t_i : t_i + t^{\text{OUT}}]$ | The future data in $\mathbf{X}^{\text{ATK}}$ at the timestamp $t_i$ |
| $\mathcal{S}$ | The set of target variables to be attacked |
| $\mathcal{T}$ | The set of all the timestamps |
| $\mathcal{T}^{\text{ATK}}$ | The set of the timestamps to be attacked |
| $f$ | The forecasting model |
| $f_s$ | The surrogate forecasting model |
| $f_g$ | The trigger generator |

# B Descriptions of Datasets

Table 6: Statistics of datasets

| Datasets | Time span | The number of variables |
|---|---|---|
| PEMS03 | 26208 | 358 |
| PEMS04 | 16992 | 307 |
| PEMS08 | 17856 | 170 |
| Weather | 52696 | 21 |
| ETTm1 | 69680 | 7 |

In this paper, we demonstrate the effectiveness of BACKTIME on five different real-world dataset, PEMS03 [63], PEMS04 [63], PEMS08 [63], Weather [2], and ETTm1 [81]. The statistics of datasets are provided in Table 6, and the detailed information is listed below.

- **PEMS datasets.** These datasets are collected by the Caltrans Performance Measurement System(PeMS) in real time every 30 seconds[4]. The traffic data are aggregated into 5-minutes intervals, which means there are 288 time steps in the traffic flow for one day. The

system has more than 39,000 detectors deployed on the highway in the major metropolitan areas in California. There are three kinds of traffic measurements contained in the raw data, including traffic flow, average speed, and average occupancy.

- **Weather dataset.** This dataset contains local climatological data for nearly 1,600 U.S. locations, 4 years from 2010 to 2013, where data points are collected every 1 hour. Each data point consists of the target value "wet bulb" and 11 climate features.

- **ETTm1 dataset.** The ETT is a crucial indicator in the electric power long-term deployment. We collected 2-year data from two separated counties in China. The ETTm1 data is collected for 15-minute-level. Each data point consists of the target value "oil temperature" and 6 power load features.

## C  Experiment Protocol

### C.1  Forecasting Models

To validate that BACKTIME is model-agnostic, we train three state-of-the-art forecasting models, including TimesNet [69], FEDformer [82], and Autoformer[9], on poisoned datasets. In the experiment, for each forecasting model, we use the default hyperparameter settings in the released code of corresponding publications [3]. We use Adam optimizer with a learning rate of $0.0002$ to update these models. These models serve as benchmarks for evaluating the effectiveness and versatility of BACKTIME across different model architectures. More details of these models are provided as follows.

- **TimesNet [69].** This model discovers the multi-periodicity adaptively and extract the complex temporal variations from transformed 2D tensors by a parameter-efficient inception block.

- **FEDformer [82].** This model utilizes Fourier transform to develop a frequency enhanced Transformer, aiming to enhance the performance and efficiency of Transformer for long-term prediction.

- **Autoformer [9].** By employing Auto-Correlation mechanism based on the series periodicity, this model conducts the dependencies discovery and representation aggregation at the sub-series level, demonstrating progressive decomposition capacities for complex time series.

### C.2  Training Settings of BACKTIME

We utilize FEDformer [82] as the surrogate forecasting model for trigger generation. Concerning BACKTIME, we adopt $t^{\text{TGR}} = 4$, $t^{\text{PTN}} = 7$ and $t^{\text{BEF}} = 6$, with the temporal injection rate $\alpha_{\text{T}}$ being $0.03$ and the spatial injection rate $\alpha_{\text{S}}$ being $0.3$. We further set $k = 200$, $\Delta^{\text{TGR}} = 0.2\text{std}$ and $\Delta^{\text{PTN}} = 0.4\text{std}$ for each dataset where $\text{std}$ represents the standard deviation of the training set. Moreover, we set $\lambda = 2,000$ for PEMS03, PEMS04, PEMS08, and Weather datasets, while $\lambda = 5$ for ETTm1 dataset. We use 2-layer MLP with the hidden layer of $64$ for graph structure generation in Eq. 4 and use 2-layer GCN with the hidden layer of $64$ as the backbone of our trigger generator.

### C.3  Baseline Methods

We compare BACKTIME with clean training strategy and three different trigger generation methods.

- ***Clean.*** Models will not be attacked and will be trained on clean datasets.

- ***Random.*** Timestamps for attack are randomly selected, and the trigger is generated from a uniform distribution ranging from $-\Delta^{\text{TGR}}$ to $\Delta^{\text{TGR}}$. This trigger is repeatedly used at each selected timestamp.

- ***Inverse.*** This attack method flips the dataset along the temporal dimension and further trains a "forecasting" model that forecasts the history based on future data. By using the target pattern as input, the outputs of the prediction model are chosen as the trigger. In experiments,

---

[3]https://github.com/thuml/Time-Series-Library

FEDformer [82] is used as the forecasting model. Please note that, for this attack method, the amplitude of generated triggers may exceed the trigger constraints, i.e., $\Delta^{\mathrm{TGR}}$.

- **_Manhattan._** This attack method locates time segments in the training set with the smallest Manhattan distance to the target pattern and uses the preceding time series data of those segments as triggers. Please note that, for this attack method, the amplitude of generated triggers may exceed the trigger constraints, i.e., $\Delta^{\mathrm{TGR}}$.

## D Description of Triggers and Target Patterns

Table 7: The value of triggers on the timestamp selection experiment.

| Timestamp | 1 | 2 | 3 | 4 |
|---|---|---|---|---|
| **Trigger** | -0.05 | 0.05 | -0.05 | 0.05 |

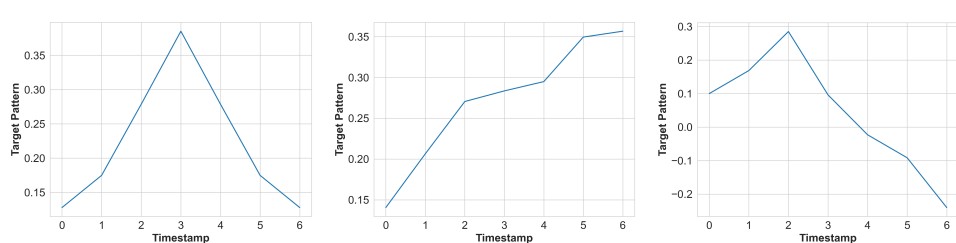

(a) The cone-shaped target pattern.

(b) The target pattern with a upward trend.

(c) The target pattern with an up and down shape

Figure 4: The shapes of all the target patterns we evaluated in this paper.

In Section 3.2.1, we implement a simple and weak backdoor attack for identifying the properties of timestamps that are more vulnerable to attack. As for the setting of this backdoor attack, we use a shape-fixed trigger, whose data are listed in Table 7, and a cone-shaped target pattern, whose data are shown in Figure 4. For each timestamp group, we will inject this trigger and target pattern into every timestamp within the group, thus poisoning $10\%$ timestamps in the training set. The experimental results show that timestamps where a clean model performs poorly are more susceptible to backdoor attacks.

In Section 4, we validate the effectiveness of BACKTIME on three different shapes of the target patterns. These target patterns will be inserted into datasets with data standardization, and the specific shapes of the three target patterns are shown in Figure 4. The endpoints of the three target patterns are equal to, higher than, or lower than their starting points, respectively. Intuitively, flipping the target patterns vertically should yield similar effects. Therefore, this paper focuses on target patterns that exhibit an upward trend after the starting point. Under these three target patterns, this paper demonstrates that BackTime can effectively attack various target patterns in MTS forecasting.

## E Main Experiment Results

The full results for BACKTIME and baselines with the cone-shaped target pattern are provided in Table 8. From the results, we can observe that BACKTIME can continuously decrease $\mathrm{MAE_A}$ to a low degree under any model architecture and any dataset. On PEMS03, PEMS04, and Weather datasets, BACKTIME surpass all the attack baselines, achieving the lowest $\mathrm{MAE_A}$ across all the model architectures. It strongly demonstrates the effectiveness and versatility of BACKTIME.

Table 8: Main results of backdoor attack on MTS forecasting. For all the metrics, the lower the better. Bold font indicates the best performance for the attack effectiveness.

| Dataset | Models | Clean | | Random | | Inverse | | Manhattan | | BACKTIME | |
|---|---|---|---|---|---|---|---|---|---|---|---|
| | | $MAE_C$ | $MAE_A$ | $MAE_C$ | $MAE_A$ | $MAE_C$ | $MAE_A$ | $MAE_C$ | $MAE_A$ | $MAE_C$ | $MAE_A$ |
| PEMS03 | TimesNet | 20.00 | 28.63 | 20.92 | 29.30 | 20.03 | 26.62 | 19.89 | 26.33 | 21.23 | **20.83** |
| | FEDformer | 15.78 | 39.86 | 16.14 | 15.70 | 16.18 | 16.05 | 16.42 | 17.10 | 16.34 | **14.05** |
| | Autoformer | 16.03 | 38.38 | 17.09 | 20.98 | 17.23 | 20.55 | 16.75 | 22.13 | 17.12 | **17.68** |
| | Average | 17.27 | 35.62 | 18.05 | 21.99 | 17.81 | 21.07 | 17.69 | 21.85 | 18.23 | **17.52** |
| PEMS04 | TimesNet | 22.95 | 51.43 | 23.94 | 46.27 | 24.47 | 33.56 | 23.91 | 38.55 | 24.43 | **25.66** |
| | FEDformer | 28.83 | 44.75 | 17.95 | **16.67** | 21.23 | 20.52 | 21.37 | 25.89 | 21.51 | 25.92 |
| | Autoformer | 21.24 | 44.28 | 22.62 | 27.08 | 22.12 | **24.43** | 22.80 | 28.41 | 21.86 | 26.94 |
| | Average | 24.34 | 46.82 | 21.50 | 30.01 | 22.61 | **26.17** | 22.69 | 30.95 | 22.60 | **26.17** |
| PEMS08 | TimesNet | 21.66 | 55.66 | 22.21 | 39.18 | 22.68 | 37.20 | 22.61 | 31.00 | 23.17 | **27.60** |
| | FEDformer | 17.87 | 27.83 | 18.08 | 31.35 | 18.29 | 28.03 | 19.07 | 18.47 | 17.70 | **16.59** |
| | Autoformer | 18.38 | 38.48 | 19.13 | 33.54 | 19.30 | 25.95 | 19.44 | 23.94 | 18.13 | **20.24** |
| | Average | 19.30 | 40.66 | 19.81 | 34.69 | 20.09 | 30.39 | 20.37 | 24.47 | 19.67 | **21.48** |
| Weather | TimesNet | 17.95 | 91.86 | 21.73 | 18.39 | 18.74 | 46.71 | 24.87 | 44.84 | 8.38 | **14.97** |
| | FEDformer | 9.83 | 97.07 | 11.13 | 16.88 | 9.85 | 77.74 | 10.35 | 95.50 | 8.64 | **15.87** |
| | Autoformer | 10.47 | 94.36 | 10.73 | 36.01 | 12.42 | 72.23 | 11.41 | 81.31 | 8.28 | **15.63** |
| | Average | 12.75 | 94.43 | 14.53 | 23.76 | 13.67 | 65.56 | 15.54 | 73.88 | 8.43 | **15.49** |
| ETTm1 | TimesNet | 1.25 | 2.50 | 1.31 | 1.67 | 1.33 | 1.49 | 1.31 | 1.63 | 1.20 | **1.45** |
| | FEDformer | 1.19 | 2.55 | 1.21 | 1.56 | 1.27 | 1.70 | 1.20 | 1.87 | 1.10 | **1.35** |
| | Autoformer | 1.32 | 2.68 | 1.32 | 1.54 | 1.36 | **1.41** | 1.34 | 1.97 | 1.12 | 1.42 |
| | Average | 1.25 | 2.58 | 1.28 | 1.59 | 1.32 | 1.53 | 1.28 | 1.82 | 1.14 | **1.41** |

