# OpenReview forum: "BackTime: Backdoor Attacks on Multivariate Time Series Forecasting"
_NeurIPS.cc/2024/Conference — NeurIPS 2024 spotlight_

### Official Review · Reviewer_Bpy4 · 2024-06-29

**Soundness:** 3
**Presentation:** 3
**Contribution:** 3
**Rating:** 7
**Confidence:** 2

**Summary:**

The paper titled "BACKTIME: Backdoor Attacks on Multivariate Time Series Forecasting" introduces a novel method called BACKTIME, aimed at exploring the robustness of multivariate time series (MTS) forecasting models against backdoor attacks. BACKTIME enables an attacker to subtly manipulate predictions by injecting stealthy triggers into the MTS data. Through a bi-level optimization process using a graph neural network (GNN)-based trigger generator, the method identifies vulnerable timestamps and crafts effective, covert triggers. Comprehensive experiments across various datasets and cutting-edge MTS forecasting models demonstrate the efficacy, flexibility, and stealth of BACKTIME attacks.

**Strengths:**

- The BACKTIME method offers a fresh perspective on backdoor attacks in MTS forecasting, demonstrating the feasibility and impact of such attacks.
- The experimental setup is comprehensive and includes multiple datasets and state-of-the-art forecasting models, showcasing the method's versatility.
- The paper provides clear explanations of the bi-level optimization process and the role of the GNN-based trigger generator, facilitating understanding of the attack mechanism.
- The research highlights the need for robustness in MTS forecasting models, particularly in high-stake scenarios where malicious attacks could have serious consequences.

**Weaknesses:**

- The paper does not adequately address potential limitations or discuss the robustness of the method against countermeasures. A more detailed analysis of the limitations and how they might affect the practical applicability of BACKTIME would strengthen the paper.
- The potential negative societal impacts of such attacks are not sufficiently explored. Given the sensitive nature of MTS forecasting in areas like climate, epidemiology, and finance, a deeper discussion on the broader implications is warranted.

**Questions:**

- What are the potential limitations of the proposed method, especially regarding its generalizability to other types of time series data or forecasting models?

**Limitations:**

- The authors have not adequately addressed the limitations of their work, nor have they discussed the potential negative societal impacts of BACKTIME.

---

> ### Author Rebuttal · Authors · 2024-08-06
>
> > **Q1. The paper does not adequately address potential limitations or discuss the robustness of the method against countermeasures. A more detailed analysis of the limitations and how they might affect the practical applicability of BACKTIME would strengthen the paper.**
>
> Thanks for the reviewer's valuable comments. We acknowledge that BackTime indeed has certain limitations.
>
> Firstly, it is challenging to leverage BackTime on time series imputation tasks. To achieve an effective backdoor attack, BackTime concatenates the trigger and the target pattern sequentially to establish a strong temporal association. This association is the foundation of BackTime's attack efficiency. However, in time series imputation, deep learning models can infer based on data both before and after the missing values. Thus, the models may not rely heavily on the data preceding the missing values, which breaks the basic assumptions of BackTime. Hence, BackTime may not be able to implement an effective attack in this scenario.
>
> Secondly, BackTime may struggle with datasets that contain missing data. BackTime predicts the target pattern only when the trigger is included within the inputs. Therefore, BackTime poses a requirement for the completeness of triggers. If the trigger itself is incomplete, the attacked deep learning model may fail to recognize the existence of the triggers, leading to an ineffective backdoor attack.
>
> Thirdly, the success of backdoor attacks relies on the redundancy of the learning capacity of deep learning models. In other words, deep learning models can complete specific tasks using only a subset of neurons, while the remaining neurons respond weakly or not at all. Thus, backdoor attacks aim to manipulate these weakly responding neurons to execute the attack without compromising the model's effectiveness. Consequently, backdoor attacks are typically more effective on deeper models with larger parameter sizes, such as TimesNet, Autoformer, and FEDformer in our paper. If the victim chooses a simple model for time series forecasting, such as MLP, the effectiveness of BackTime might deteriorate.
>
> Another limitation of this work is that it does not offer a solution for defense. To address this gap, we have outlined some preliminary ideas on how to defend against BackTime. First, there may be a frequency difference between the generated triggers and the target pattern compared to the real data. Hence, detecting the distribution drift in the frequency domain could be a promising approach. Additionally, the generated triggers might not exhibit the same rich diversity as real data. Consequently, in the feature space, the (trigger, target pattern) pair might cluster closely with each other, making detection via clustering algorithms feasible.
>
> > **Q2. The potential negative societal impacts of such attacks are not sufficiently explored. Given the sensitive nature of MTS forecasting in areas like climate, epidemiology, and finance, a deeper discussion on the broader implications is warranted.**
>
> Thank you for your concern regarding the potential societal impacts of BackTime. As a backdoor attack method, the misuse of BackTime indeed poses the risk of negative social consequences.
>
> Specifically, in the financial markets, attackers could exploit BackTime to inject stealthy backdoors into stock price prediction models, enabling market manipulation and illicit profit gains. This could lead to increased market volatility and eroded investor confidence. In public transportation, as mentioned in lines 27 - 32 of our paper, BackTime could cause transportation prediction models to output incorrect results. This could negatively impact traffic signal control and route planning, resulting in increased traffic congestion and accidents. In the healthcare field, medical monitoring systems often rely on time series data, such as ECG datasets. A backdoor attack could cause data-driven healthcare models to produce erroneous diagnostic results, potentially affecting treatment plans and endangering patient health.
>
> Given the potential risks of BackTime, we present several ways to mitigate its negative societal impacts. From the perspective of data collection, researchers should be aware of the risk of backdoor injection into publicly available time series datasets. Therefore, when using public datasets for training, it is recommended to rigorously inspect and cleanse the data to minimize the impact from malicious data injection. We have provided some possible data detection methods in Q1. Due to the character limit, we do not provide a detailed description here. From the perspective of model training, researchers can enhance model robustness. For example, by employing adversarial training strategies, researchers can reduce the model's sensitivity to specific perturbations, thus enabling the model to resist backdoor trigger samples. From the perspective of model deployment, researchers can establish real-time monitoring systems to detect anomalies in system inputs and outputs. Effective anomaly detection algorithms can help promptly identify potential attacks.
>
> Once again, thank you for your insightful comments and suggestions. We will incorporate our response to both Q1 and Q2 in the revised version.

---

> ### Comment · Reviewer_Bpy4 · 2024-08-13
>
> Thank you for the detailed rebuttal. The clarification of BACKTIME's limitations and the discussion on societal impacts are appreciated. Addressing these aspects will indeed strengthen the paper. Please ensure these points are integrated into the manuscript for the next review. Continue to consider potential defenses and broader implications for various applications. My concern has been addressed. I will raise my score to accept.

---

> > ### Author Response · Authors · 2024-08-13
> > **Response to Reviewer Bpy4**
> >
> > We want to express our sincere thanks for your recognition of our work. Answering your insightful comments helps us improve the quality of our work. In the revised version of our paper, we promise to include discussions on potential defenses and broader implications. We are confident that these additions will enrich our contribution and provide a more comprehensive understanding of the impact and context of our work.

---

### Official Review · Reviewer_PmgC · 2024-07-10

**Soundness:** 4
**Presentation:** 3
**Contribution:** 4
**Rating:** 8
**Confidence:** 4

**Summary:**

This paper proposed a backdoor attach method for multivariate time series forecasting, where only a few works focus on this topic. The injected stealthy triggers are interesting and effective in destroying raw data.

**Strengths:**

The originality of this paper is solid, since this is the first work to consider the backdoor attack in MTS forecasting. The quality and clarity of this paper are also good for understanding. The significance of this paper is relatively well-clarified.

**Weaknesses:**

1. The Trigger Generation and Bi-level Optimization are not clearly presented. Please extend the description of trigger generation and effectiveness.
2. The experiments need to be expanded, and the advantages of the proposed method are not well validated.
3. Numerically, the authors could consider comparing their method with more baselines. There are some studies on backdoor attacks for forecasting models.

**Questions:**

1. The Trigger Generation and Bi-level Optimization are not clearly presented. Please extend the description of trigger generation and effectiveness.
2. The experiments need to be expanded, and the advantages of the proposed method are not well validated.
3. Numerically, the authors could consider comparing their method with more baselines. There are some studies on backdoor attacks for forecasting models.

**Limitations:**

1. What is the technical drawback of the proposed method? E.g., attack efficiency and hidden effect.
2. Are there any initial ideas to be able to look forward from a defense perspective?

---

> ### Author Rebuttal · Authors · 2024-08-06
>
> > **Q1. The Trigger Generation and Bi-level Optimization are not clearly presented. Please extend the description of trigger generation and effectiveness.**
>
> Thank you for your attention to the trigger generation and bi-level optimization. These two components are indeed core parts of BackTime. However, due to the page limit, we were unable to provide a detailed description in the original version. We are more than willing to elaborate on these two components in the revised version.
>
> Regarding trigger generation, for the sake of stealthiness, we aim to generate triggers that are similar to the data preceding the injected triggers. Therefore, we extract the time series data before the injected triggers, denoted as $\mathbf{X}^{ATK}[t_i - t^{BEF} - t^{TGR}:t_i - t^{TGR}]$ in Eq. (5), and use it as the input to the trigger generator (a GCN). The GCN effectively generates the trigger by integrating the time series data before triggers and the correlation graph. However, as mentioned in the paper, the amplitude of the GCN's output could be large, which poses a significant challenge to the convergence of the training process. To address this, we apply the tanh function to scale the output, and utilize the scaled outputs as the final triggers as shown in Eq. (6).
>
> During the training process, the trigger generator is updated adaptively in an end-to-end manner. Specifically, the trigger generator injects the generated trigger into the poisoned dataset $\mathbf{X}^{ATK}$, and the surrogate model $f_s$ is leveraged to evaluate and update the trigger. Intuitively, an effective trigger should induce $f_s$ to predict the future as the target pattern. Thus, after freezing the parameters of $f_s$, we use  $l_{tgr}$ in Eq. (10) as the loss function to update the trigger generator, aiming to reduce the difference between the predictions of $f_s$ and the target pattern.
>
> > **Q2. The advantages of the proposed method are not well validated, and the authors could consider comparing BackTime with more baselines. There are some studies on backdoor attacks for forecasting models.**
>
> We agree with the reviewer in that the baselines in this paper are heuristic and MAE may not fully evaluate our method. Meanwhile, we would like to emphasize the superiority of BackTime from two perspectives.
>
> First, on four datasets (PEMS03, PEMS08, Weather, and ETT), our method reduces $MAE_A$ by 50.8\%, 52.64\%, 83.52\%, and 45.40\% compared to clean training, respectively. Even when compared with the best baseline in this paper, BackTime also reduces $MAE_A$ by 16.84\%, 12.26\%, 34.80\%, and 7.84\%. These results demonstrate that BackTime effectively implements a backdoor attack on multivariate time series (MTS).
>
> Second, we would like to highlight that our work is the **first** to focus on backdoor attacks in MTS forecasting and formally define a backdoor attack framework for time series forecasting. As such, finding suitable backdoor attack methods for MTS forecasting as baselines is challenging. To our best knowledge, there are not existing baselines in the literature that directly apply to our setting. Therefore, we designed a few heuristic methods as the baselines.
>
> To better address reviewer's concerns,  we have provided additional classic adversarial attack methods, such as FGSM and PGD, as baselines. Specifically, we generated adversarial perturbations to induce the model to predict the target pattern. The experimental results are as follows. From the table, PGD and FGSM not only failed to implement effective backdoor attacks but also significantly reduced the model's forecasting performance. One of the reasons is that the generated adversarial perturbations exhibit significant difference from the real data, hence impeding models' learning on clean features. This highlights the necessity for meticulously designed triggers in backdoor attacks on time series forecasting and randomly chosen perturbations highly likely fail to implement effective attacks.
>
> If there is any specific baseline the reviewer would like to let us include, we would be more than happy to include them in our evaluation.
>
> |          | $MAE_C$ | $RMSE_C$ | $MAE_A$ | $RMSE_A$ |
> |----------|---------|----------|---------|----------|
> | Clean    | 20.00   | 34.18    | 28.63   | 46.69    |
> | FGSM     | 110.04  | 151.71   | 91.13   | 149.76   |
> | PGD      | 108.86  | 150.43   | 90.01   | 147.07   |
> | BackTime | 21.23   | 35.22    | 20.83   | 30.94    |
>
> > **Q3. What is the technical drawback of the proposed method? E.g., attack efficiency and hidden effect.**
>
> Thank you for your attention to the technical drawbacks and limitations of BackTime. We acknowledge that BackTime may not exhibit good attack efficiency in certain application scenarios. However, due to the character limit, we kindly ask the reviewers to refer to our response to the reviewer Bpy4's Q1 for a more comprehensive discussion.
>
> > **Q4. Are there any initial ideas to be able to look forward from a defense perspective?**
>
> Thank you for your attention to potential defenses against backdoor attacks in time series data. Backdoor defense is indeed an important and promising future direction. While it will require a separate study to develop a full solution, we do have some initial ideas for backdoor defenses. First, there may be a frequency difference between the generated triggers and the target pattern compared to the real data. Hence, detecting the distribution drift in the frequency domain could be a promising approach. Additionally, the generated triggers might not exhibit the same rich diversity as real data. Consequently, in the feature space, the (trigger, target pattern) pair might cluster closely with each other, making detection via clustering algorithms feasible.

---

> > ### Comment · Reviewer_PmgC · 2024-08-12
> >
> > Thanks for the authors' efforts in the detailed response. My concern has been addressed. I will raise my score to strong accept.

---

> > > ### Author Response · Authors · 2024-08-13
> > > **Response to Reviewer PmgC**
> > >
> > > We sincerely thank you for your appreciation of our work. Your comments have been invaluable in guiding us in improving the quality of our research. In the revised version of the paper, we promise to include detailed descriptions of the trigger generation and bi-level optimization processes, as requested. Additionally, we will incorporate a discussion of potential defenses against the proposed attack. We believe that these enhancements will significantly strengthen our work and provide a more comprehensive understanding of its contributions.

---

### Official Review · Reviewer_mt9q · 2024-07-14

**Soundness:** 2
**Presentation:** 4
**Contribution:** 3
**Rating:** 6
**Confidence:** 2

**Summary:**

This paper primarily discusses how to conduct a backdoor attack on the multivariate time series forecasting task, proposing a two-stage training approach. The core idea is to identify timestamps with significant differences in MAE of the Clean model and Poisoned model on poisoned data points. Simultaneously, the trigger generation function is learned by minimizing the MAE loss of the model's prediction of the target pattern with triggers as input.

**Strengths:**

1. This is a quite interesting and important research topic. However, the example provided by the author regarding traffic prediction is not convincing enough. Traffic prediction data is collected and trained by private systems. If one can already access the system to inject toxic data, there would be simpler and more effective methods than backdoor attacks. If the author can provide some scenarios where the time series data points are provided by third-party systems, it would better illustrate the necessity of this paper.
2. The author's writing is clear and easy to follow.
3. The author has proposed a framework that can be applied to various time series prediction backbones and has conducted a comprehensive evaluation of effectiveness.

**Weaknesses:**

1. The baseline methods in the paper are heuristic, and the worse MAE of these methods compared to the proposed method in the paper does not effectively illustrate the issue.
2. The types of anomalies illustrated in the paper are relatively simple.

**Questions:**

1. In Equation 3, according to the paragraph from lines 189-190, the clean loss should be the MAE using the clean samples to predict the ground truth, and the attack loss should be the MAE using the poisoned samples to predict the target pattern, right? However, in Equation 3, both MAE losses target $X^{ATK}_{ti,f}$. Here, clarification from the author is requested.
2. The training of the trigger generation network depends on initially manually generating some attack samples. Does the quality of these initial samples significantly affect the following training process?
3. The two-stage training is very unstable, which the author mentioned in the paper, but there is no detailed description in the text. It would be helpful to understand the rationality of the model design if the author could provide the training loss change curve.

**Limitations:**

The author has discussed in the paper how to prevent the injected data points from having too large a magnitude and how to address training instability issues.

---

> ### Author Rebuttal · Authors · 2024-08-06
>
> > **Q1. The baseline methods in the paper are heuristic, and the worse MAE of these methods compared to the proposed method in the paper does not effectively illustrate the issue.**
>
> Thank you for your concern on the baselines in our papers. We agree with the reviewer that the baselines in this paper are heuristic and MAE may not fully evaluate our method. However, due to the character limit, we kindly ask the reviewer to refer to our response to the reviewer Pmgc's Q2 for a more comprehensive discussion.
>
> > **Q2. The types of anomalies illustrated in the paper are relatively simple.**
>
> Thank you very much for your valuable suggestions. As the reviewer correctly pointed out, the main goal of this work is to demonstrate the feasibility of backdoor attacks on multivariate time series, and therefore, we did not focus on complex target patterns. However, we would like to emphasize that BackTime can effectively perform attacks, even with complex target patterns.
>
> To illustrate this, we have conducted additional experiments on the PEMS03 dataset, attacking the TimesNet model using three different complex target patterns. Since it is challenging to design complex target patterns when the pattern length is too short, we set the pattern length to 17 in the new experiments instead of 7 as mentioned in the paper. The target patterns are provided at Figure 1 with the format of PDF on the global rebuttal.
>
> The experimental results are presented in the following table. As shown in the table, $MAE_A$ and $RMSE_A$ reach a quite low value, even lower than $MAE_C$ and $RMSE_C$. It demonstrates that BackTime successfully completes the attack even with complex patterns, demonstrating its superiority. Once again, thank you for your insightful comments and suggestions. They have been incredibly helpful in further improving our work.
>
> |           | $MAE_C$ | $RMSE_C$ | $MAE_A$ | $RMSE_A$ |
> |-----------|--------|---------|--------|---------|
> | Pattern 1 | 22.42  | 36.88   | 20.41  | 29.80   |
> | Pattern 2 | 22.85  | 37.35   | 20.35  | 29.78   |
> | Pattern 3 | 22.47  | 36.97   | 20.37  | 29.70   |
>
> > **Q3. In Eq. (3), the clean loss should be the MAE using the clean samples to predict the ground truth, and the attack loss should be the MAE using the poisoned samples to predict the target pattern, right? However, in Eq. (3), both MAE losses target $X_{ti,f}^{ATK}$.**
>
> Thanks for your valuable comments on our notations. As you pointed out, the attack loss is used to update the trigger for the purpose of predicting the target pattern. To achieve this, we introduce $\beta(t_i)$ in the upper-level optimization to locate the target pattern within $X^{ATK}$.
>
> However, there seems to be a misunderstanding regarding the precise role of the clean loss. $X^{ATk}$, the attacked dataset, contains both clean time slices and attacked time slices. Victims may train their model on this attacked dataset $X^{ATK}$, resulting in an attacked model that carries the backdoor. Therefore, the lower-level optimization still uses $X^{ATK}$, and the clean loss here refers to the common loss function in the forecasting task, such as MAE. We believe the term "clean loss" might have caused this misunderstanding. Therefore, we will change this term to "natural loss" in the revised version to clarify its meaning.
>
> > **Q4. The training of the trigger generation network depends on initially manually generating some attack samples. Does the quality of these initial samples significantly affect the following training process?**
>
> Thank you for your attention to the trigger generation network. As mentioned in line 264 of our paper, we introduced the "adaptive trigger generation" mechanism. Thus, after end-to-end training of the trigger generator, triggers could be adaptively acquired, and there is no need for manually setting the initial attack samples.
>
> Specifically, the trigger generator injects the generated trigger into the poisoned dataset $\mathbf{X}^{ATK}$, and the surrogate model $f_s$ is leveraged to evaluate and update the trigger. Intuitively, an effective trigger should induce $f_s$ to predict the future as the target pattern. Thus, after freezing the parameters of $f_s$, we update the parameter of the trigger generator to reduce the difference between the predictions of $f_s$ and the target pattern. Once the trigger generator is well-trained, it can adaptively output effective triggers.
>
> Perhaps the current description on trigger generation has caused this misunderstanding. We will provide a more detailed explanation in the revised version.
>
> > **Q5. Could the author provide some scenarios where the time series data points are provided by third-party systems?**
>
> Thank you for your concern about the scenarios. BackTime can be leveraged in several application scenarios where the data are collected by third-party systems, beyond just traffic prediction. However, due to the character limit, we kindly ask the reviewer to refer to the second paragraph of our response to reviewer Bpy4's Q2 for a detailed discussion.
>
> > **Q6. The two-stage training is unstable, but there is no detailed description in the text. It would be helpful to understand the rationality of the model design if the author could provide the loss curve.**
>
> Thank you for your valuable comments. The two-stage training is a crucial part of BackTime's training process. Sorry for not providing detailed explanations in the original version due to the page limit. We are more than willing to include a thorough description of this process in the updated version of our paper.
>
> We have also provided a detailed explanation in response to the Reviewer Pmgc's Q1, and we hope these clarifications will resolve some confusions. Additionally, as the reviewer's request, we have provided the loss function curves in Figure 2 on the global rebuttal to help understand the two-stage training process.

---

> > ### Comment · Reviewer_mt9q · 2024-08-11
> >
> > Thank the author for the reply. The explanation regarding clean loss and trigger generation has addressed my concerns. However, the examples in response to reviewer Bpy4 has not convinced me. In both the medical and stock market, the attacks are still conducted after data collection. If one can access the data in such private system, the most effective method would be to directly tamper with the data rather than using the indirect approach of constructing toxic samples to influence the model. The current method in the paper requires training to determine what kind of attack samples to generate, which I believe is a limited contribution. I am more looking forward to seeing attack methods and defense methods that are training-free. In total, I think this paper has not yet reached the level of a score 7 (clear accept), and I maintain my current score of 6 as weak accept."

---

> > > ### Author Response · Authors · 2024-08-12
> > > **Response to Reviewer mt9q**
> > >
> > > We are very glad to know that your concerns about the clean loss and trigger generation have been addressed. We also want to thank you for the follow-up questions and are happy to share more answers.
> > >
> > > First, we would like to point out that standard backdoor attacks belong to the category of **data-poisoning** attacks, where modifications/augmentations to training data are a necessary and indispensable requirement in backdoor attacks across various domains [1,2,3,4], such that the attacked model learn the strong association between triggers and target pattern (or target class) in the training data, thereby altering its predictive behavior. Unlike traditional attacks that might rely solely on exposed data and could be short-sighted or ineffective, backdoor attacks strategically manipulate the training data to embed malicious behavior, which remains dormant during normal operations but can be triggered under specific conditions. This approach ensures that the attack is both stealthy and persistent.
> > >
> > > Furthermore, under this data-poisoning setting, backdoor attacks can be highly threatening because neural network training requires a large amount of data, which is difficult to accomplish by only using private data. Therefore, by stealthily poisoning public datasets, attackers can pose a great threat to neural networks' wide application.
> > >
> > > For example, in the healthcare field, it is hard for a single hospital to collect enough data to train a robust model. Hence, hospitals are likely to delegate to AI-related agencies, and those agencies mix the hospital-provided data with large-scale public data (e.g., scraped from the internet) to enhance model performance. By poisoning widely used public datasets, such as ERP [5] and OpenNeuro [6] datasets, backdoor attacks could pose a significant threat to the medical control systems used by hospitals.
> > >
> > > The suggestion of a training-free strategy is quite interesting but indeed presents significant challenges to attack efficiency based on the current developments in the research community. Since a training-free backdoor attack, neither has any knowledge of the architecture of models to be trained nor has access to the data during training, the trigger can only be manually set. However, time series data varies greatly in both frequency and amplitude, ranging from traffic data to weather patterns. It is exceedingly difficult for a manually designed trigger to perform an effective attack across different time series datasets. Considering this, we believe that developing a training-free backdoor attack is a highly challenging research problem, and we would be very interested in exploring this in future research.
> > >
> > > [1] Badnets: Identifying vulnerabilities in the machine learning model supply chain.
> > >
> > > [2] Backdoor attack with imperceptible input and latent modification
> > >
> > > [3] Textual backdoor attack for the text classification system.
> > >
> > > [4] A backdoor attack against lstm-based text classification systems.
> > >
> > > [5] https://erpinfo.org/erp-core
> > >
> > > [6] https://openneuro.org/

---

### Author Rebuttal · Authors · 2024-08-07

1. Figure 1 visualizes three different target patterns.
    - Experiments on the response to the reviewer mt9q’s Q2 show that BackTime successfully completes the attack even with complex target patterns.
2. Figure 2 visualizes the loss curve of the two-stage training process in BackTime.
    - We use the same color to plot the two loss ($l_{cln}$ and $l_{tgr}$) within the same epoch.
    - During each epoch of the two-stage training, we run the clean training first, as shown in the upper figure, and then update the trigger generator, as shown in the lower figure

---

### Decision · Program_Chairs · 2024-09-25

**Decision:**

Accept (spotlight)

**Comment:**

This paper proposes a backdoor attack method on multivariate time series forecasting task. It uses MAE to identify attack susceptible time frame, and then use GCN to generate an adaptive trigger. The method is trained with a bi-level optimization. Reviewers agree that the problem and the solution are novel, and the validation is sufficient.